# White Matter Microstructural Alterations in Newly Diagnosed Parkinson’s Disease: A Whole-Brain Analysis Using dMRI

**DOI:** 10.3390/brainsci12020227

**Published:** 2022-02-07

**Authors:** Jun-Yeop Kim, Jae-Hyuk Shim, Hyeon-Man Baek

**Affiliations:** 1College of Medicine, Gachon University, Incheon 21565, Korea; luckyjunyeop10@gmail.com; 2Department of Health Sciences and Technology, GAIHST, Gachon University, Incheon 21999, Korea; jaehyukshim11@gmail.com

**Keywords:** Parkinson’s disease, whole brain, diffusion MRI, quantitative anisotropy (QA)

## Abstract

Parkinson’s disease (PD) is a neurodegenerative disorder characterized by cardinal motor symptoms and other non-motor symptoms. Studies have investigated various brain areas in PD by detecting white matter alterations using diffusion magnetic resonance imaging processing techniques, which can produce diffusion metrics such as fractional anisotropy and quantitative anisotropy. In this study, we compared the quantitative anisotropy of whole brain regions throughout the subcortical and cortical areas between newly diagnosed PD patients and healthy controls. Additionally, we evaluated the correlations between the quantitative anisotropy of each region and respective neuropsychological test scores to identify the areas most affected by each neuropsychological dysfunction in PD. We found significant quantitative anisotropy differences in several subcortical structures such as the basal ganglia, limbic system, and brain stem as well as in cortical structures such as the temporal lobe, occipital lobe, and insular lobe. Additionally, we found that quantitative anisotropy of some subcortical structures such as the basal ganglia, cerebellum, and brain stem showed the highest correlations with motor dysfunction, whereas cortical structures such as the temporal lobe and occipital lobe showed the highest correlations with olfactory dysfunction in PD. Our study also showed evidence regarding potential neural compensation by revealing higher diffusion metric values in early-stage PD than in healthy controls. We anticipate that our results will improve our understanding of PD’s pathophysiology.

## 1. Introduction

Parkinson’s disease (PD) is a degenerative disorder characterized by progressive cardinal motor symptoms such as resting tremor, rigidity, bradykinesia, and postural instability. Such symptoms can be traced to the various systematic changes that occur in motor networks due to dopamine depletion, mainly in the substantia nigra pars compacta (SNpc) [1,2]. In addition to these motor symptoms, non-motor symptoms (NMS), including olfaction dysfunction, autonomic dysfunction, and REM sleep behavior disorder (RBD), also manifest in PD due to α-synuclein accumulation in specific pathways throughout the brain [3]. Many studies have suggested that NMS can be used as a diagnostic marker of prodromal and/or early PD [4,5]. Recently, diffusion magnetic resonance imaging (dMRI) has been used to diagnose early PD by assessing white matter (WM) connectivity in some brain areas related to motor symptoms as well as NMS [6].

dMRI is a unique tool that can be used to measure WM microstructure in vivo by characterizing the diffusion displacement of water molecules in white matter. Diffusion tensor imaging (DTI) has been used as a common dMRI processing technique to obtain gross fiber orientation of white matter voxels and use them to calculate quantitative diffusion metrics, such as fractional anisotropy (FA) and mean diffusivity (MD) [7]. Various studies have shown correlations between significantly high or low levels of diffusion metrics and alterations in WM microstructural integrity, such as demyelination and necrosis [8,9,10]. However, while DTI has been used in a significant number of studies, it has some limitations such as the inability to resolve complex fiber orientations such as crossing or branching patterns. To overcome such limitations, a novel approach called generalized q-sampling imaging (GQI) was developed. Unlike DTI, which is a model-based method, GQI is a model-free method based on the Fourier transform between diffusion MR signals and the underlying diffusion displacement, providing more precise directional and quantitative information regarding neural structures. Overall, GQI has shown advantages over conventional diffusion tensor approaches due to its higher sensitivity in fiber tracking and wide applicability [11]. GQI provides a new quantitative diffusion metric called quantitative anisotropy (QA), which represents the density of WM from the calculations of water diffusivity in specific directions and has shown better accuracy in reconstructing images compared with other conventional metrics such as FA [12]. Due to the multiple advantages mentioned above, many studies have used GQI to investigate the pathological mechanisms of WM microstructures. For instance, QA alterations showed significant correlations with various pathological changes such as the peritumoral infiltration of gliomas and demyelination in radiation-induced brain injuries [13,14].

Many studies have used dMRI to compare diffusion metrics between PD patients and healthy controls (HCs) to observe whether the degeneration of neuronal microstructure can be observed in the form of significant diffusion metric changes throughout various areas of the brain. Most studies focused on specific brain areas, such as the substantia nigra (SN) and basal ganglia, which are well known to be affected in PD [15,16]. While there have been some studies that have investigated multiple brain areas other than the areas mentioned above [17], there is only a small number of studies focusing on the entire brain area using diffusion metrics. Several recent dMRI analysis studies have shifted from using conventional metrics to using QA; however, most of these studies investigated the differences in specific WM tracts rather than in whole brain areas between HCs and PD patients [18,19,20]. In addition, studies using diffusion metrics, such as FA and QA, have provided conflicting results. Although decreased diffusion metrics have been frequently reported in PD [21,22,23], there are several occasions where results showed higher FA and QA in PD than in HC, which were commonly interpreted as neural compensation that occur in the early stages of neurodegeneration [17,24,25]. In this regard, a systemic review published in 2020 suggested that regions other than the hallmark regions of PD, such as SN, should be investigated more extensively and coherently using advanced dMRI processing techniques [26].

In this study, we extensively compared the QA of multiple brain regions between PD and HC. We compared not only the regions affected directly by dopaminergic neuronal loss, but also some scarcely investigated regions. We delineated various structures and classified them into larger parts such as the basal ganglia, limbic system, cerebellum, thalamus, brain stem, corpus callosum, frontal lobe, parietal lobe, occipital lobe, temporal lobe, and insular lobe. In addition, we analyzed the correlations between the QAs of various brain areas and neuropsychological test scores in PD to determine the impact of WM degeneration in each brain area on several neuropsychological dysfunctions. We anticipate that this study can improve the current understanding of PD neuropathology through dMRI processing of several areas rarely touched upon in PD studies.

## 2. Materials and Methods

### 2.1. Participants

A total of 44 HCs and 44 PD participants in this study were recruited from the Parkinson’s Progression Markers Initiative (PPMI) open-access database (www.ppmi-info.org/data, accessed on 1 September 2021). Each PD patient was assessed for PD using part III of the Movement Disorder Society-sponsored revision of the United Parkinson’s Disease Rating Scale (MDS-UPDRS III), dopamine transporter SPECT scans to observe the dopaminergic neurons, and manifested motor symptoms such as bradykinesia and resting tremor. Assessment of PD was performed before any patients were administered PD medication, which could interfere with PD symptom testing. All participants in this study tested negative for neurological disorders other than PD. The group demographics and clinical characteristics are shown in Table 1. All participants provided written informed consent to share their unidentified clinical data with the investigators.

### 2.2. MRI Data Acquisition

The MRI data of the HC and PD participants used in this study were obtained from the PPMI database (www.ppmi-info.org/data, accessed on 1 September 2021). Diffusion MRI images were acquired using standard protocols on 3T scanners at approximately 11 PPMI imaging sites. T1-weighted images were acquired using a 3D T1-weighted MPRAGE sequence (echo time (TE) = 90 ms, repetition time (TR) = 2300 ms, 1 mm^3^ resolution), and DTI images were acquired using a 2D single-shot echo-planar DTI sequence (TE = 88 ms, TR = 900 ms, 2 mm^3^ resolution, 72 slices, flip angle = 90°, 64 gradient directions, b-value = 1000 s/mm^2^). More details regarding MRI sequence information can be found in the PPMI MRI technical operations manual (www.ppmi-info.org, accessed on 1 September 2021).

### 2.3. Image Processing

Each diffusion-weighted MRI image was preprocessed through a series of MRtrix3 image-processing programs, which involved denoising, Gibbs ringing removal, motion and distortion correction, bias field correction, and resampling to a 1 mm^3^ isotropic resolution [27,28]. The brain structures of each subject were segmented through nonlinear spatial normalization of the FreeSurfer segmentation atlas [29] from the MNI152 template space to each subject’s respective space using DSI Studio [11]. The QA diffusion measure of each brain structure was obtained using DSI Studio statistics [11].

### 2.4. Classification of Brain Structures

Each segmented structure was classified into a larger part of the brain. The following are the areas of the brain and the individual structures that comprise them: basal ganglia (caudate, putamen, accumbens area, pallidum, and ventral diencephalon); limbic system (hippocampus, amygdala, cingulate gyrus, orbital frontal gyrus, insular gyrus, parahippocampal gyrus, and subcallosal gyrus); cerebellum, thalamus, brain stem, corpus callosum, frontal lobe (precentral gyrus, superior frontal gyrus, middle frontal gyrus, inferior frontal gyrus, orbital gyrus, rectus gyrus, frontomarginal cortex, and transverse frontopolar cortex); parietal lobe (postcentral gyrus, superior parietal lobule, inferior parietal lobule, and precuneus); occipital lobe (cuneus, lingual gyrus, fusiform gyrus, superior occipital gyrus, middle occipital gyrus, inferior occipital cortex, and occipital pole); temporal lobe (superior temporal gyrus, middle temporal gyrus, inferior temporal gyrus, and temporal pole); and insular lobe.

### 2.5. Statistical Analysis

All statistical analyses were performed using IBM SPSS Statistics for Windows (version 22.0; IBM Corp., Armonk, NY, USA). Demographic, clinical, and neuropsychological data of the participants were analyzed. Within the PD group, QA values of each structure were compared between the most and least affected side using the paired *t*-test. All QA values of each individual segment of the brain were compared between HC and PD groups using the independent sample *t*-test. The Benjamini–Hochberg procedure was used to correct type 1 errors for multiple comparisons, and a false discovery rate (FDR) level of 0.005 was applied. Pearson’s correlation was used to determine the correlation coefficients between the QA values of each brain region and corresponding neuropsychological test scores (significance at *p* < 0.05).

## 3. Results

Table 1 presents the demographics and clinical characteristics of the HC and PD participants, which were compared for significant differences using the independent sample *t*-test. The “dominant side” means the side most affected at PD symptom onset, and this information was investigated due to the asymmetric characteristic in most newly diagnosed PD [30]. The following scores showed significant differences (*p* < 0.05) between HC and PD participants: the H & Y scale, an indicator of PD progression; the MDS-UPDRS III score, an indicator of movement disorders; the UPSIT score, an indicator of olfaction functions; the SCOPA-AUT score, an indicator of autonomic symptoms in PD; and the RBDSQ score, an indicator of sleep disorders. The other indices showed no significant differences (*p* > 0.05) between the HC and PD groups.

The information regarding the segmented brain regions from the PPMI database was classified by anatomical or functional criteria. Figure 1 shows each classified area of the brain, which is the main subject we investigated.

The paired *t*-test was used to compare the QAs of each structure between the most affected and the least affected side. We defined most affected side as the contralateral side of the “dominant side” mentioned above, while the least affected side was defined as the opposite side of the most affected side. As a result of the analysis, no regions showed significant QA differences between the most and the least affected side within PD (*p* > 0.05).

The independent sample *t*-test was used to compare the QAs of subcortical structures between HCs and PD patients, as shown in Table 2. With post FDR correction significance set to 0.005, PD patients showed significantly higher QAs in the basal ganglia, striatum, limbic system, cerebellum, and brain stem compared to HC individuals. Significances of QA differences were seen in both sides no matter how much the side was affected. Figure 2 shows a graph of the QA differences in the subcortical structures between the HC and PD groups.

The independent sample *t*-test was also used to compare the QAs of cortical structures between the HC and PD groups, as shown in Table 3. With post FDR correction significance set to 0.005, PD patients had significantly higher QAs in the occipital lobe, temporal lobe, and insular lobe compared to HC individuals. Significances of QA differences were seen in both sides no matter how much the side was affected. Figure 2 shows a graph of the QA differences in the cortical structures between the HC and PD groups.

Correlations between the neuropsychological test (MDS-UPDRS III, UPSIT, SCOPA-AUT, and RBDSQ) scores and QAs of each brain region (basal ganglia, limbic system, cerebellum, thalamus, brain stem, corpus callosum, frontal lobe, parietal lobe, occipital lobe, temporal lobe, and insular lobe) were evaluated by calculating correlation coefficients. We conducted this correlation analysis to find out whether each symptom of PD is related to the alterations of QA, a diffusion measure that has been shown to represent WM density in specific brain regions affected in PD. We selected the four neuropsychological tests mentioned above based on the studies that showed motor and some non-motor symptoms (olfactory dysfunction, autonomic disturbance, and sleep disturbance) appear in early-stage PD [1,5]. The results of our analysis presented in Table 1 also show significant differences between the HC and PD groups in the four tests mentioned above (*p* < 0.05). Additionally, we conducted an analysis of QA between brain regions of the most affected side and the least affected side and found no significant differences. The results are shown in Figure 3. The MDS-UPDRS III scores were significantly positively correlated with each area mentioned above (*p* < 0.05). Among them, the brain stem (r = 0.399), cerebellum (r = 0.393), and basal ganglia (r = 0.392) are the areas which showed the highest correlation coefficients. The UPSIT scores were significantly negatively correlated with every area mentioned above (*p* < 0.05). Among them, the temporal lobe (r = −0.321) and occipital lobe (r = −0.313) are the areas which showed the highest correlation coefficients in absolute values. The SCOPA-AUT scores were significantly positively correlated with the basal ganglia alone (r = 0.255). The RBDSQ showed no significant correlation with any of the regions mentioned above.

## 4. Discussion

In our study, we utilized diffusion MRI images to compare the QAs of the segmented structures throughout the whole brain. These structures included not only specific areas influenced by the dopaminergic neurons, such as the basal ganglia, but also other brain areas that have not been investigated thoroughly using dMRI techniques. Additionally, we investigated the correlations between the QAs of the brain areas and corresponding neuropsychological test scores. Our results revealed that the basal ganglia, limbic system, cerebellum, brain stem in subcortical areas, occipital lobe, temporal lobe, and insular lobe in cortical areas showed significant differences in QAs between the HC and PD groups when the FDR was set to 0.005. In the correlation analysis, the QAs of the brain stem, cerebellum, and basal ganglia showed the highest correlation coefficients with MDS-UPDRS III scores, and the QAs of the temporal lobe and occipital lobe showed the highest correlation coefficients in absolute values with UPSIT scores.

A significant number of studies on PD have investigated group differences in diffusion metrics such as FA and QA in brain structures, especially in the basal ganglia, where the death of dopaminergic neurons occurs [15,16]. The substantia nigra (SN) is the target structure that has been investigated using various diffusion metrics. However, there have been controversial results in the SN between early PD patients and controls. Some studies found neuronal degeneration and a reduced FA of SN in early PD patients [21,23,31], whereas other studies found no significant differences or even higher values of FA of SN in early PD patients when compared to controls [22,32]. Besides SN, other subcortical structures such as the putamen and cerebellum, which are involved in dopaminergic pathways, have also been investigated. Although there are some conflicting results [17,33], many studies have shown that early PD patients display FA increases in several subcortical areas, such as the putamen [24,34]. Recently, many studies have attempted to compare QA values in subcortical areas between HCs and PD patients. Studies have found that prodromal or earlier-stage PD patients showed higher QAs compared to HCs or later-stage PD patients in several WM tracts involved in subcortical areas [18,25,35]. These results are in line with the significantly higher QAs in early PD patients compared to HCs that were observed in the specific brain areas considered in our study, including the basal ganglia, cerebellum, and brain stem. These results could provide evidence of compensatory brain plasticity in early PD. Compensatory brain plasticity or neural compensation is a term that explains the dissociation between brain pathology and behavioral alterations occurring in prodromal or early stages of neurodegeneration, such as Alzheimer’s disease [36]. Our findings strengthen the hypothesis of compensatory brain plasticity in very early phase PD, as mentioned in previous studies [18,25].

Although subcortical areas have been the target in most studies using DTI, some studies have attempted to find significant results in cortical areas in PD. Structures related to olfaction have often been chosen as the investigation target in PD because olfactory dysfunction is one of the earliest clinical symptoms of PD, even preceding motor symptoms [37]. The olfactory system is an area that includes the piriform cortex and parahippocampal gyrus; it also connects to various areas, including the limbic system structures (amygdala and hippocampus) [38,39]. Previous studies have found differences in FA and QA in the areas or tracts related to the olfactory system between HCs and PD patients [18,35,40]. Our results were similar to those of previous studies in that we showed significantly higher QA values in the temporal lobe, where the olfactory cortex is located, and in the insular lobe, which is known to have a specific role in olfactory processing [41], in early PD patients compared to HCs. Our study also found a difference in QA between the HC and PD groups in the occipital lobe. This could be the result of connectivity alterations in some WM tracts such as the fronto-occipital fasciculus or longitudinal fasciculus, as shown in previous studies [20]. Otherwise, this might be the result of alterations in WM distribution in specific occipital regions such as the optic radiation [19,42], which could be related to visual dysfunction in early PD patients [43]. However, further studies are required to interpret this result more precisely. Higher QAs in PD patients compared to HCs are thought to be due to neural compensation appearing in these regions in the early stage of PD [18,25]. QAs of the frontal lobe and parietal lobe showed no significant differences between the HC and PD groups, which might be the result of only selecting early-stage PD cases as the study subjects, in whom widespread microstructural changes do not occur [25,31,44].

Motor deficits are typical symptoms of PD due to the degeneration of nigrostriatal tracts, which have been shown in many DTI-derived metrics [45]. In our study, higher correlation coefficients between the QAs of the basal ganglia, cerebellum, and brain stem and MDS-UPDRS III scores showed that the microstructure of the regions where nigrostriatal tracts pass through or are well known to be related to motor function were altered the most, and these changes contribute to motor dysfunction in early-stage PD. Olfactory dysfunction is a typical early non-motor symptom in PD [36] and is known to be related to the olfactory system, which is located in the temporal lobes [32,39,46]. In line with this knowledge, the QAs of the temporal lobes showed higher correlations with the UPSIT scores than other regions. This result confirms the previous result that pathological changes in areas of the olfactory system other than the olfactory bulb are in charge of olfactory dysfunction in PD [37]. In addition, the result of a lower correlation coefficient in areas where the nigrostriatal tract passes with UPSIT scores could be interpreted using previous results which revealed that odor identification deficits in PD are not explained by nigrostriatal dopaminergic denervation [47,48]. The QAs of the occipital lobes also showed higher correlations with the UPSIT scores. A previous study that investigated the association between olfactory dysfunction and brain microstructures in prodromal PD showed that some fibers, such as the fronto-occipital fasciculus and longitudinal fasciculus, which are connected to the occipital lobes, are significantly associated with UPSIT scores in PD [20]. Our results could be interpreted in this way, where alterations of some tracts passing through the occipital lobes are associated with olfactory dysfunction. However, further studies are required to interpret this result more precisely. MDS-UPDRS III and UPSIT scores were significantly correlated with the QAs of almost every brain area, while SCOPA-AUT and RBDSQ scores only showed significant correlations with a few brain regions, if any, even though some studies found a significant relationship between autonomic or sleep disorders and WM connectivity in PD [49,50]. These findings suggest that motor deficits and olfactory dysfunction could be typical biomarkers that appear earlier than autonomic or sleep disorders and could be correlated with widespread microstructural alterations of the brain in early PD.

Although we conducted this research with careful considerations, there are some limitations. The participants were only recruited from the PPMI database, and the number of participants used for this study may limit the generalizability of this study. It might be useful to explore whether the results of this study are reproducible in further studies. There also may be limitations caused by not dividing the PD patients into subgroups based on the severity of the disease. WM alterations can potentially be significantly different based on the severity of the disease within a single early PD group. By comparing PD subgroups, it would be possible to determine whether the severity or duration would be correlated with WM alterations in PD. One limitation regarding our study is the utilization of data using outdated PPMI protocols. According to previous research, conventional DTI scalar measures such as FA and MD are affected by different acquisition factors such as b-value, resolution, and gradient directions [51]. For example, a study showed that a higher number of gradient directions increased FA while reducing MD [52]. Although we could not find any studies investigating how QA values are affected by different DTI acquisition schemes, we predict that QA would show a similar pattern to FA. Another major limitation is the interpretation of the QAs in the occipital lobe. Although we suggested some hypotheses to interpret the QA differences between HC and PD and the correlations between the QAs of the occipital lobe and UPSIT scores, further studies are needed to explain these results more accurately. However, we believe our study is the first to evaluate whole brain structures using an advanced dMRI processing technique to compare QA as a diffusion metric between HC and PD and to correlate the QAs of PD with several neuropsychological test scores. These findings could provide valuable evidence for the pathophysiology of PD and the applicability of dMRI processing techniques in PD.

## 5. Conclusions

In this study, we utilized a dMRI processing technique called GQI to compare QA as a diffusion metric between HCs and PD patients throughout the whole brain region. We were able to find significant QA differences in not only the typical pathologic areas of PD, such as the basal ganglia and brain stem, but also in less-studied areas such as the limbic system, temporal lobe, and occipital lobe. Higher QA values in PD suggest that neural compensatory mechanisms occur in early-stage PD, similar to the results of previous studies on prodromal or early-stage PD. In addition to these results, we found that some subcortical structures such as the basal ganglia, cerebellum, and brain stem are highly correlated with motor dysfunction in PD, while some cortical structures such as the temporal lobe and occipital lobe are highly correlated with olfactory dysfunction in PD. We believe that these results could broaden our understanding of PD. Future studies on PD will be necessary to reproduce and verify our findings.

## Figures and Tables

**Figure 1 brainsci-12-00227-f001:**
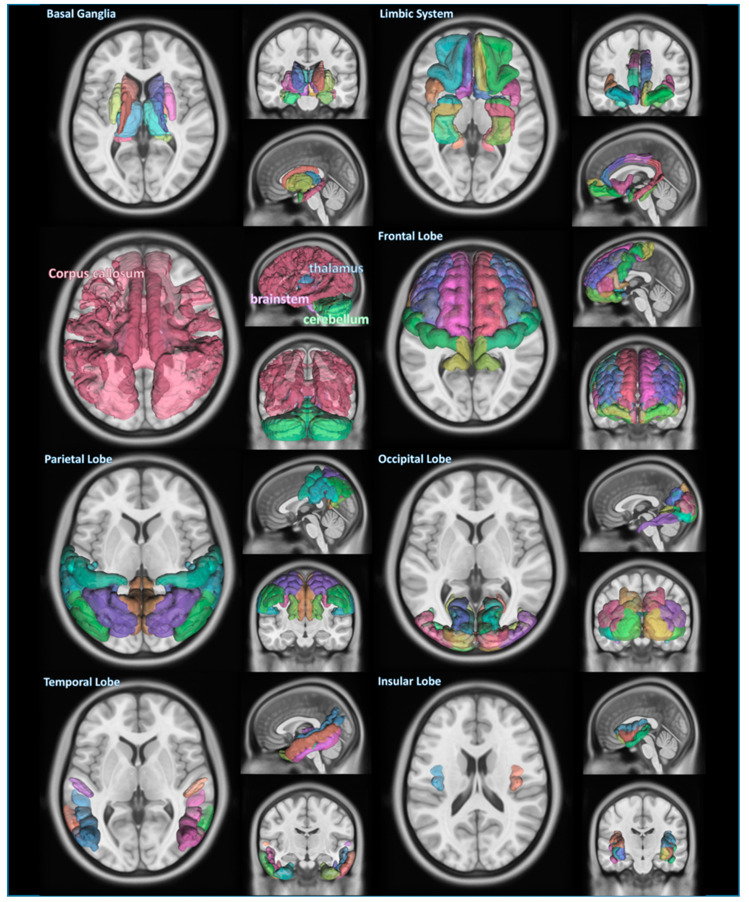
Segmentation of several areas of brain in Parkinson’s disease. The areas composed of segmented structures of a PD patient recruited in this study are basal ganglia, limbic system, cerebellum, thalamus, brain stem, corpus callosum, frontal lobe, parietal lobe, occipital lobe, temporal lobe, and insular lobe, which are overlaid on top of MNI templates.

**Figure 2 brainsci-12-00227-f002:**
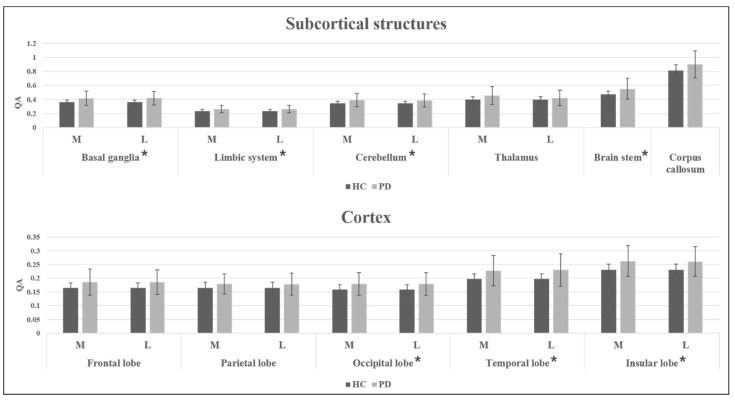
QA differences in subcortical and cortical structures between HC and PD groups. “M” and “L” represent the “most affected side” and “least affected side” in PD. Error bars represent standard errors. Structures with asterisk represent the regions with significant difference after FDR correction.

**Figure 3 brainsci-12-00227-f003:**
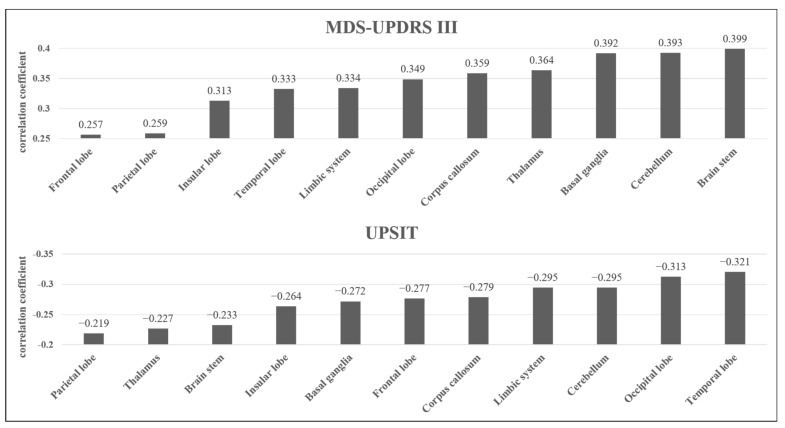
Correlation coefficients of each brain area with MDS-UPDRS III and UPSIT scores.

**Table 1 brainsci-12-00227-t001:** Group characteristics of PD and HC.

Group	HC (*n* = 44)	PD (*n* = 44)	*p*-Value
Age (mean ± SD)	60.4 ± 9.6	58.3 ± 9.3	0.316
Sex (male/female)	27/17	27/17	-
Dominant side (left/right)	-	23/21	-
Education years (mean ± SD)	16.2 ± 2.9	15.3 ± 3.1	0.186
Age onset in years (mean ± SD)	-	56.6 ± 9.7	-
Duration of disease in months (mean ± SD)	-	6.1 ± 6.4	-
H & Y scale (mean ± SD)	0.0 ± 0.0	1.5 ± 0.5	<0.001
MDS-UPDRS III score (mean ± SD)	0.7 ± 1.1	19.7 ± 9.1	<0.001
UPSIT score (mean ± SD)	33.3 ± 4.8	23.7 ± 6.9	<0.001
SCOPA-AUT score (mean ± SD)	5.4 ± 2.7	8.7 ± 5.7	<0.001
RBDSQ score (mean ± SD)	2.6 ± 2.0	3.6 ± 2.0	0.017
GDS score (mean ± SD)	1.2 ± 2.5	1.8 ± 1.7	0.154
MoCA score (mean ± SD)	28.4 ± 1.1	28.0 ± 1.7	0.188

HC, healthy controls; PD, Parkinson’s disease; SD, standard deviation; H & Y, Hoehn and Yahr; MDS-UPDRS III, Movement Disorder Society-sponsored revision of the Unified Parkinson’s Disease Rating Scale; UPSIT, University of Pennsylvania Smell Identification Test; SCOPA-AUT, Scale for Outcomes in Parkinson’s disease—Autonomic; RBDSQ, REM Sleep Behavior Disorder Screening Questionnaire; GDS, Global Deterioration Scale; MoCA, Montreal Cognitive Assessment Test Scoring.

**Table 2 brainsci-12-00227-t002:** QA differences in the subcortical structures between HC and PD groups.

Region	Subregion	HC QA (Mean ± SD)	PD QA (Mean ± SD)	*p*1 Value	*p*2 Value
Basal ganglia		0.361 ± 0.031	0.417 ± 0.103 (M)	0.001	**0.002**
			0.418 ± 0.095 (L)	<0.001	**<0.001**
	Striatum	0.329 ± 0.030	0.385 ± 0.086 (M)	<0.001	**<0.001**
			0.383 ± 0.093 (L)	<0.001	**<0.001**
Limbic system		0.234 ± 0.020	0.263 ± 0.054 (M)	0.002	**0.002**
			0.263 ± 0.053 (L)	0.001	**0.002**
	Cingulate gyrus	0.209 ± 0.018	0.228 ± 0.043 (M)	0.008	0.004
			0.229 ± 0.047 (L)	0.011	0.005
Cerebellum		0.343 ± 0.029	0.389 ± 0.094 (M)	0.003	**0.003**
			0.386 ± 0.092 (L)	0.002	**0.003**
Thalamus		0.398 ± 0.038	0.454 ± 0.130 (M)	0.008	0.004
			0.422 ± 0.111 (L)	0.003	0.003
Brain stem		0.474 ± 0.044	0.551 ± 0.149	0.002	**0.003**
Corpus callosum		0.811 ± 0.081	0.901 ± 0.193	0.006	0.004

Average QA values of the subcortical structures. “M” and “L” in PD QA section represent the “most affected side” and “least affected side”, respectively. Significant differences and *p*-values were calculated using independent sample *t*-tests (*p*1 value), with *p*-values adjusted for multiple corrections using FDR correction (*p*2 value). Bolded values represent QAs with significant differences after FDR correction (FDR = 0.005). QA: quantitative anisotropy.

**Table 3 brainsci-12-00227-t003:** QA differences in the cortical structures between HC and PD groups.

Region	HC QA (Mean ± SD)	PD QA (Mean ± SD)	*p*1 Value	*p*2 Value
Frontal lobe	0.164 ± 0.018	0.186 ± 0.048 (M)	0.007	0.004
		0.186 ± 0.045 (L)	0.004	0.004
Parietal lobe	0.164 ± 0.022	0.179 ± 0.037 (M)	0.030	0.005
		0.178 ± 0.040 (L)	0.043	0.005
Occipital lobe	0.158 ± 0.018	0.179 ± 0.041 (M)	0.002	**0.003**
		0.179 ± 0.041 (L)	0.003	**0.003**
Temporal lobe	0.197 ± 0.018	0.227 ± 0.055 (M)	0.001	**0.001**
		0.230 ± 0.059 (L)	<0.001	**<0.001**
Insular lobe	0.231 ± 0.021	0.262 ± 0.056 (M)	<0.001	**0.001**
		0.261 ± 0.055 (L)	0.002	**0.002**

Average QA values of cortical structures. “M” and “L” in PD QA section represent the “most affected side” and “least affected side”, respectively. Significant differences and *p*-values were calculated using independent sample *t*-tests (*p*1 value), with *p*-values adjusted for multiple corrections using FDR correction (*p*2 value). Bolded values represent QAs with significant differences after FDR correction (FDR = 0.005). QA: quantitative anisotropy.

## Data Availability

Data used in the preparation of this article were obtained from the Parkinson’s Progression Markers Initiative (PPMI) open-access database (www.ppmi-info.org/data, accessed on 1 March 2021).

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
