# Peer review of "White Matter Microstructural Alterations in Newly Diagnosed Parkinson’s Disease: A Whole-Brain Analysis Using dMRI"

_brainsci, 2022, doi:10.3390/brainsci12020227_

Round 1

Reviewer 1 Report

In the study „White matter microstructural alterations in newly diagnosed Parkinson’s disease: A whole-brain analysis using dMRI“ by Kim et al. the authors applied GQI modeling and whole-brain analyses on PPMI-derived DTI datasets. In general, the manuscript is well written and can be considered for publication. I have only a few suggestions that may help to improve the manuscript:

The authors should provide more in-depth information regarding the GQI approach. The passage in the Introduction has been written rather technically. It would substantially improve the readability if this aspect could be rephrased for non-MRI physicists.

It would be interesting to know, whether QA alterations have already been linked to any histopathological changes in brain tissues. For now, biological interpretability is lacking.

I may have missed that, but are there any particular reasons why only 2 x 44 PPMI participants have been included in the analysis?

The DTI acquisition protocol of the PPMI study is rather outdated. The authors should discuss this limitation and maybe state how QA values are affected by different DTI acquisition schemes.

It would make more sense to perform correlation analyses of motor symptoms (as expressed by the UPDRS-III) with brain structures of the clinically more affected side.

Is there any particular reason why the SN has been spared from analyses? If so (e.g., because of improper segmentation issues), the authors should give a statement.

Are there any hypothesis-driven correlation analyses between QA metrics and clinical scores? At present, these analyses appear to be somewhat arbitrary. This section should be rephrased to avoid the impression of data-fishing.

The manuscript needs some in-depth English editing. There are several grammatical errors and cumbersome phrasings.

Author Response

Thank you for your kind review. We made some modifications to the manuscript according to your suggestions.

Reviewer 2 Report

  Novel, interesting, well-designed and well-presented work.

  I have one major comment which, if adequately addressed, will increase the impact of the study:

MAJOR COMMENT

  Newly diagnosed sporadic Parkinson’s disease is usually asymmetric in its symptomatology as well as in its imaging findings (at least DAT). In the manuscript the issue of asymmetry is addressed only in Table 1, where it is reported that the “dominant side” was the left in 23 and the right in 21 patients, without clarifying the meaning of the term. Does “dominant” mean the side of symptom presence/predominance?

    It will be extremely interesting (and very easy with data already available) to assess dMRI asymmetries by performing:

(a) PAIRED t-tests comparing “most affected” vs “less or not affected side” (not “left” vs “right”) in PD.

      Paired testing is expected to reveal much more significant differences

(b) t-tests comparing HC with “most affected” side in PD

    This comparison is expected to yield much more significant differences than the “left” vs “right” comparison which is shown in Tables 2 & 3 and Figure 2.

(c) t-tests comparing HC with “less affected” side in PD

    It will be interesting to see whether the “less affected” side in PD has QAs significantly different than HC.

  The comparisons suggested are expected to yield straightforward results in part because, as can be seen in Tables 2 & 3, none of the structures appear to show a significant left-right QA difference in HC (neither in PD).

MINOR COMMENTS

Introduction, Line 29

“…progressive and degenerative cardinal motor symptoms…”

The term degenerative cannot be used to characterize symptoms. The disease is degenerative, not the symptoms.

Materials and Methods, Line 86

“Dopamine transporter SPECT scans were used to observe the dopaminergic neurons and display motor symptoms such as bradykinesia and resting tremor.”

The two parts of the sentence are not connected. Something is missing.

Author Response

(The authors gave the same response as above.)

Round 2

Reviewer 1 Report

The authors substantially improved their manuscript. I have no further concerns and recommend the acceptance of the publication in its current form.